# A mathematical model of fibrinogen-mediated erythrocyte–erythrocyte adhesion

Catarina S. Lopes [1,8], Juliana Curty[2,8], Filomena A. Carvalho[1], A. Hernández-Machado[3,4,5,6], Koji Kinoshita [7], Nuno C. Santos [1✉] & Rui D. M. Travasso [2✉]

Erythrocytes are deformable cells that undergo progressive biophysical and biochemical changes affecting the normal blood flow. Fibrinogen, one of the most abundant plasma proteins, is a primary determinant for changes in haemorheological properties, and a major independent risk factor for cardiovascular diseases. In this study, the adhesion between human erythrocytes is measured by atomic force microscopy (AFM) and its effect observed by micropipette aspiration technique, in the absence and presence of fibrinogen. These experimental data are then used in the development of a mathematical model to examine the biomedical relevant interaction between two erythrocytes. Our designed mathematical model is able to explore the erythrocyte–erythrocyte adhesion forces and changes in erythrocyte morphology. AFM erythrocyte–erythrocyte adhesion data show that the work and detachment force necessary to overcome the adhesion between two erythrocytes increase in the presence of fibrinogen. The changes in erythrocyte morphology, the strong cell-cell adhesion and the slow separation of the two cells are successfully followed in the mathematical simulation. Erythrocyte-erythrocyte adhesion forces and energies are quantified and matched with experimental data. The changes observed on erythrocyte–erythrocyte interactions may give important insights about the pathophysiological relevance of fibrinogen and erythrocyte aggregation in hindering microcirculatory blood flow.

[1] Instituto de Medicina Molecular, Faculdade de Medicina, Universidade de Lisboa, Lisbon, Portugal. [2] CFisUC, Department of Physics, University of Coimbra, Coimbra, Portugal. [3] Departament de Física de la Matèria Condensada, Facultat de Física, Universitat de Barcelona, Barcelona, Spain. [4] Centre de Recerca Matemàtica, Bellaterra, Spain. [5] Barcelona Graduate School of Mathematics (BGSMath), Barcelona, Spain. [6] Institute of Nanoscience and Nanotechnology (IN2UB), Universitat de Barcelona, Barcelona, Spain. [7] Department of Molecular Medicine, University of Southern Denmark, Odense, Denmark. [8] These authors contributed equally: Catarina S. Lopes, Juliana Curty. ✉email: nsantos@fm.ul.pt; ruit@uc.pt

Erythrocytes are adaptable cells that undergo progressive physical and chemical changes during the inflammatory pathophysiology in the vascular system[1]. Biochemical changes can be noticed in erythrocytes, such as the decrease on cell volume with cell aging. Among other factors, this is presumably due to the loss of potassium and the loss of membrane patches by microvesiculation, resulting in an increase on cell density[1,2]. Erythrocyte deformability is an important factor affecting the high shear blood viscosity. Under normal conditions, erythrocytes deform sufficiently to cross blood vessels of different calibres[3]. The erythrocytes' membrane, forming a complex with the cytoskeleton, may also exhibit plastic changes under some pathological circumstances and can be permanently deformed by excessive shear forces[4]. Mechanical properties of erythrocytes strongly affect blood flow. For example, low erythrocyte deformability increases blood viscosity and flow resistance, which lead to several cardiovascular and cerebrovascular conditions[5].

Interestingly, erythrocyte adhesion mediated by receptors is crucial in haematological diseases such as sickle cell anaemia[6], malaria[7] and type 2 diabetes mellitus[8]. Blood platelets also drive erythrocyte aggregation, leading to clot formation. This mechanism is crucial in wound healing or thrombus-induced strokes. In healthy donors, erythrocytes have a tendency to form aggregates called rouleaux that look similar to a stack of coins when no flow is applied[9]. These rouleaux can be separated into smaller units by means of a shear flow, giving rise to the shear thinning behaviour of blood.

Despite erythrocytes being a major component of thrombi[9], in blood, fibrinogen (one of the most abundant plasma proteins) is a primary determinant of changes in haemorheological properties, and these alterations exacerbate the complications in peripheral blood circulation during cardiovascular pathologies[10]. Fibrinogen is involved in the coagulation cascade, it has a crucial role in blood clotting and has been identified as a major independent risk factor for cardiovascular diseases[11–20]. An increase in fibrinogen plasma concentration promotes erythrocyte aggregation[21]. Using atomic force microscopy (AFM)-based force spectroscopy, we previously identified the specific receptor for fibrinogen on the erythrocyte membrane[12,17,22]. The simultaneous binding of fibrinogen to such receptors on two different erythrocytes may transiently bridge these cells, increasing blood viscosity at the microcirculatory level, impairing a proper blood flow and increasing vascular risk.

Erythrocyte–erythrocyte adhesion has been measured by different approaches, including AFM[12,17,22]. Micropipette aspiration, developed by Evan Evans in the 1980s[23–25], may also be used. It has been extensively used to measure the mechanical properties of erythrocytes membranes, including membrane elastic modulus and membrane viscosity[26,27], to analyse the mechanical properties of bilayers[28], the mechanical and adhesive properties of single cells on substrates or cell–cell adhesion[29–31]. Cell morphology (cell shape and spreading) can be visualised during experiments and it is also an initial indication of the propensity for adhesion[32].

Phase-field models have been used by the physical community to study domain formation and growth in phase separating non-equilibrium systems[33–35]. These models are particularly useful in describing the dynamical properties of the interface or membrane separating different domains. In biology, phase-field models have been extensively used to describe tumour growth[36,37], vessel growth[38–41], as well as cell deformation and dynamics[42–46].

Mathematical models based on the minimisation of energy functionals are efficient in determining cell morphology evolution. Phase-field models[33–35], in particular, are especially useful in describing the dynamical properties of the interface or membrane separating different domains. In biology, they have been extensively used to describe tumour growth[36,37], vessel growth[38–41,47], multiscale modelling of thrombus in blood vessels[48], interactions between cells and the extracellular matrix[46], and cell deformation and motility[42–44,46]. Interestingly, the model proposed by Nonomura (2012)[42,45] can be applied to investigate cellular morphology alterations due to cell–cell interaction, e.g., in epithelial tissues, in tumour growth, or as a consequence of differential adhesion between cell types.

In this study, we present a theoretical model for the evolution of two erythrocytes' mechanical interaction while they are pushed against each other, and then pulled apart. We used the experimental assays (AFM and micropipette aspiration technique) to measure the erythrocytes typical adhesion force between two cells (Fig. 1). Afterwards, we used these values to parameterise the model and to explore in more detail the cell–cell interactions as a function of time.

In the Results and Discussion section, we describe the observations from the micropipette aspiration assays (Fig. 1a) and the AFM measurements (Fig. 1b) carried out to explore the adhesion between erythrocytes in the presence of fibrinogen. We also show how our mathematical model is able to explore the cell–cell interaction forces and cell morphology alterations during the cell–cell interaction. In the Conclusions section, we discuss the results of this work and draw the final conclusions. We describe the experimental setup and the mathematical model in the "Methods" section.

## Results and discussion

**Erythrocyte–erythrocyte adhesion evaluation by micropipette experiments.** Cell–cell adhesion studies were performed with and without the addition of fibrinogen (1.0 mg/mL), evaluating the binding between two erythrocytes from healthy donors (Fig. 2). The total amount of erythrocyte pairs analysed were 32 and 34 with and without fibrinogen addition, respectively. For each erythrocyte pair, we obtained $16 \pm 5$ adhesion image sequences, with a 15 s cycle each, and determined if adhesion between the cells was observed. Often it can be visualised that this adhesion leads to a pull of the membrane of the erythrocyte of the left side (static micropipette) when the erythrocyte from right micropipette is being moved away from it. Moreover, in those situations it can be observed that the adhesion between the two erythrocytes forces the erythrocyte on the right to bend to the left as it is being pulled away. In Fig. 2 we present seven representative images from three cycles to illustrate cell–cell interaction in events where no adhesion is observed (with and without fibrinogen addition, Fig. 2a, b, respectively), and in an event where adhesion between the cells is present (Fig. 2c). Events with changes in erythrocyte shape were often absent in the sequences without fibrinogen, while they were frequently observed when fibrinogen was added to the experiment (Fig. 2d).

**Erythrocyte–erythrocyte adhesion evaluation by AFM experiments.** Cell–cell adhesion was also evaluated by AFM-based force spectroscopy experiments (Fig. 3). It is observed that the work necessary to overcome the adhesion of two erythrocytes is lower in the absence of fibrinogen than in the presence of 1.0 mg/mL of fibrinogen ($0.63 \pm 0.12$ fJ vs. $1.45 \pm 0.34$ fJ, respectively; $p = 0.030$; Fig. 3a). The erythrocyte–erythrocyte maximum detachment force was also quantified in the same conditions (Fig. 3b). Without fibrinogen, the maximum detachment force between erythrocytes was $168.7 \pm 33.9$ pN, which was lower than in the presence of fibrinogen 1.0 mg/mL ($330.5 \pm 65.2$ pN; $p = 0.030$; Fig. 3b). Thus, fibrinogen promotes a stronger erythrocyte adhesion.

**Simulation results.** The simulation of the micropipette assay was then carried out using the mathematical model described by Eq. (2) (see Methods section). Initially, the model was used to equilibrate two erythrocytes with the target volume and surface area from Table 1.

## a - Micropipette Aspiration

## b - Atomic Force Microscopy

**Fig. 1 Schematic experimental setup representation of the erythrocyte–erythrocyte adhesion experiments carried out using micropipette aspiration and atomic force microscopy.** Using the micropipette aspiration technique (**a**), a negative pressure (suction) is applied to catch an erythrocyte in each micropipette. One of these micropipettes (depicted as the one on the left side of each panel from I to IV) is kept static. The piezo-driven micropipette, with an attached erythrocyte, approaches the other erythrocyte on the static micropipette, until a certain contact force is reached (I). Then, during the retraction of the right side pipette, cell–cell adhesion may occur, leading to the pulling of the surface of the erythrocyte from the left side when the erythrocyte from right side micropipette is being moved away from it (II and III), until the complete erythrocyte–erythrocyte detachment is reached (IV). Using atomic force microscopy (AFM) (**b**), the tipless cantilever with an attached erythrocyte is approached to other erythrocyte on the substrate, until a certain contact force is reached (I). Afterwards, during retraction, cell–cell adhesion may occur, causing the cantilever to adhere to the sample up to a distance beyond the initial contact point (II). When the applied force overcomes the cell–cell interaction forces, the cantilever pulls of sharply (III), before totally breaking the cell–cell contact and complete the erythrocyte detachment (IV).

According to the minimisation of the Canham-Helfrich energy, the ratio $\nu = V_{\text{target}} / (\frac{4}{3}\pi(\frac{S_{\text{target}}}{4\pi})^{3/2})$ between the enclosed volume, $V_{\text{target}}$, and the volume of a sphere with the same surface area, $S_{\text{target}}$, as the vesicle determines its equilibrium shape[49,50]. For $\nu \gtrsim 0.65$, the vesicle presents a prolate shape, approaching a sphere when $\nu \sim 1$. For $\nu \lesssim 0.55$, the vesicle free-energy is minimised by bending into a stomatocyte shape. Oblate shapes, on the other hand, are obtained at the vicinity of $\nu \sim 0.60$. The values used for the erythrocytes surface area and volume lead to $\nu = 0.614$, and therefore we obtain the characteristic erythrocyte oblate morphology.

In this simulation, the two erythrocytes were moved towards each other with relative advection velocity $2v = 1.14\,\mu\text{m/s}$ (see Table 1) until they collide. Then, the erythrocytes were moved backwards with the same absolute velocity. During the simulation, the forces on the erythrocytes were measured and the alteration in erythrocyte morphology was characterised.

We started by simulating an aligned collision between the two erythrocytes. In this collision, the two erythrocytes are parallel to each other and their relative velocity is parallel to the line that connects the erythrocytes' centres and to the $z$ axis. The value for the adhesion coefficient in this simulation was set to $\eta = 8.0 \times 10^{-12}$ J/m to simulate attractive forces on the range of 300 pN, as observed in the AFM assays (Fig. 3b and Fig. 4c). This adhesion coefficient corresponds to an adhesion energy between two membranes of $\frac{9\sqrt{2}}{35\epsilon}\eta = 17.1\,\text{aJ/}\mu\text{m}^2$ (this expression is obtained directly from the integration of the energy adhesion term between two interfaces in their equilibrium state). As the two erythrocytes collide, the contact area between them is on the range of 50 to 100 $\mu\text{m}^2$, and so the simulated energy of adhesion between the two cells will be on the order of 1 to 2 fJ, similarly to the values observed experimentally (Fig. 3a).

Figure 4 presents the timelapse of the collision between the two erythrocytes as they approach and then recede, as a 2D mid-slice of erythrocytes profiles (Fig. 4a), and as a 3D representation (Fig. 4b). The total force exerted on the right erythrocyte along the three axes is plotted in Fig. 4c as a function of time. In this frontal collision, due to the symmetry of the collision, we observe a zero total force in directions $x$ and $y$. On the other hand, along the collision axis, the erythrocytes initially undergo an attractive force, as they approach each other (first minimum in Fig. 4c). This attractive force leads to an alteration in cell morphology: in the attraction region the cells flatten to maximise the region of contact (Fig. 4a, panel 3). As the cells are pushed further against each other, the hard core repulsion term, regulated by $\gamma$ in Eq. (2), is responsible for the appearance of a strong repulsion force between the two erythrocytes (maximum in Fig. 4c), which rapidly increases as the cells get closer. At $t = 1.18$ s in the simulation, the motion of the cell is reversed and the repulsion force rapidly decreases. As the two erythrocytes break apart, the adhesion force works to pull them together. We observe a rapid increase in adhesion force until the cells are sufficiently apart and the force between them then decreases to zero.

The model of Eq. (2) may be used to explore the force profiles for different collisions between two erythrocytes. In Fig. 5a, b, we plot the timelapse for an off-centred collision. In this case, the two erythrocytes are still parallel to each other, but their centres are not aligned, and they approach with an offset distance $d$ (the length of the projection along the $y$ axis of the vector connecting the two cell centres).

The force applied in the cell along the velocity direction follows the same trend as in the aligned collision (Fig. 4c and Fig. 5c). As the cells approach, there is an adhesion force that may lead to small changes in cell morphology. Then, as the cells are pushed against each other, the hard core repulsion force increases rapidly. Finally as the cells are pushed away from each other, the adhesion force reaches its maximum before decreasing to zero, when the cells are far from each other. Notably, for higher values of the

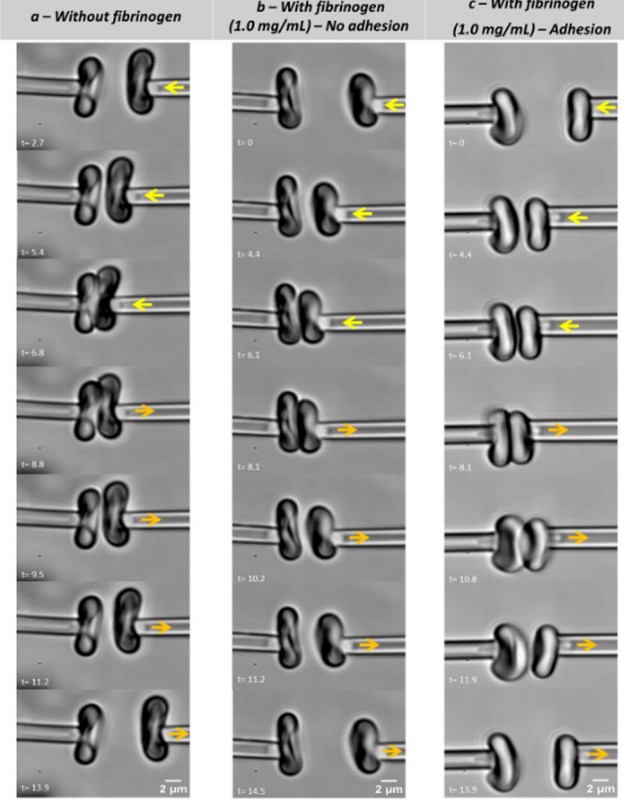

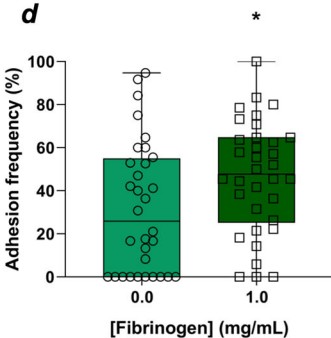

**Fig. 2 Sequence of microscopy images of the micropipette aspiration assay and measured erythrocyte–erythrocyte adhesion frequency.** Images at different stages of the erythrocyte–erythrocyte adhesion in the absence of fibrinogen (**a**) and in the presence of 1.0 mg/mL of fibrinogen (**b, c**). A negative pressure (20 Pa) was applied to catch an erythrocyte in each micropipette (left side pipette is kept static). The arrows indicate the direction of movement of the erythrocyte on the right. Erythrocyte-erythrocyte adhesion was continuously observed on a digital video for subsequent analysis with LabVIEW. It is possible to observe that in the presence of fibrinogen there was a higher adhesion between erythrocytes, leading to a pulling of the membrane of the erythrocyte of the left side (on the static micropipette) when the erythrocyte from right micropipette is being moved away from it. **d** Erythrocyte-erythrocyte adhesion frequency. The represented values are the percentage of the adhesion between two erythrocytes from healthy donors. The frequency of adhesion between two erythrocytes in the presence of fibrinogen 1.0 mg/mL is higher than on its absence (46.7 ± 26.0% vs. 32.0 ± 30.4%, respectively; $p = 0.04$). The total number of erythrocyte pairs analysed, obtained from five healthy donors, were $n = 32$ and $n = 34$, with and without fibrinogen addition, respectively. For each erythrocyte pair, we obtained 16 ± 5 adhesion image sequences. Experimental values are presented together with box-and-whiskers plot with the median value.

offset, the overlap between the two cells is smaller and the force magnitude in the $z$ direction decreases.

In the direction of the offset (the $y$ direction), there is a non-zero force applied in the cell. As the erythrocyte on the right approaches the other one, it is pushed vertically for the two shapes to better fit each other. In this way, the magnitude of the vertical force depends non-monotonously on the offset. This force is still the sum of the cell–cell attraction force and the cell–cell hard core repulsion. After the initial contact, instead of flattening out, as in the aligned collision, in the collision with an offset the cell membranes adjust to better maximise the contact area. In this way, the thicker part of the erythrocyte attempts to "dock" into the middle dimple of the other erythrocyte. The forces measured in the $y$ direction are the ones driving this push on the cell on the right to adapt its shape to the cell on the left.

This "docking" of the cells implies that they are pushed closer together without overlapping than in the aligned collision described above (Fig. 4), and therefore the peak of hard core repulsion between the cells is smaller than in the aligned scenario (Fig. 5c), but it also has a small component in the negative $y$ direction for the offset values between $d = 1.02\,\mu\text{m}$ and $d = 4.08\,\mu\text{m}$, which can be clearly identified in the force profiles (Fig. 5d).

More drastic cell deformations can be observed for larger values of adhesion coefficient. For $\eta = 2.7 \times 10^{-11}$ J/m, for example, we observe a large alteration in cell morphology when the cells are being pulled apart (see Fig. 6 for an offset $d = 4.08\,\mu\text{m}$), akin to the largest changes in erythrocyte morphology observed in our micropipette assay (e.g. in Fig. 2c). In this scenario, the cells adhere strongly to one another. When they are pulled back, the sections of the cells which are not adhered move easily backwards, while the adhered section takes a longer time to separate (Fig. 6a, b). Similarly to the previous cases, as the cells approach, we initially observe the adhesion force between them, and the cells maximise locally the area of contact (Fig. 6c). As the cells are further pushed against each other, the repulsion force prevents the cell–cell overlap. In this simulation the contact region is small (a consequence of the large offset), but a local maximum in the force between the two cells due to the repulsion force at $t = 1.18$ s can still be observed. As the cells are then pulled apart, we observe a peak in adhesion. As they separate from each other, the force profile indicates a slow decrease in the adhesion force between the cells, consequence of a slow and progressive separation process (Fig. 6c). These strong adhesion and slow separation lead to dramatic changes in erythrocyte morphology (Fig. 6a, b).

## Conclusion

In this work we presented a coupled experimental and theoretical approach to analyse erythrocyte adhesion forces during cell–cell contact. The AFM experimental measurements of the adhesion between cells enabled the parameterisation of a novel mathematical model, which can be used to explore, in finer detail, the process of cell–cell adhesion mediated by fibrinogen.

One of the major achievements of this work was to study fibrinogen-mediated erythrocyte aggregation using micropipette manipulation and atomic force microscopy. Erythrocyte-erythrocyte adhesion was measured by both techniques in the absence and presence of fibrinogen. Erythrocytes with higher adhesion (high values of cell–cell detachment work and force) in the presence of fibrinogen could be associated with a higher risk of thromboembolic events and a worse clinical prognosis. We previously reported that the deformability, measured in an ektacytometer in conditions mimicking low calibre vessels, of the erythrocyte of chronic heart failure patients is lower than for healthy donors[17]. Using AFM to assess erythrocytes from these patients, we showed that higher forces are needed to break the

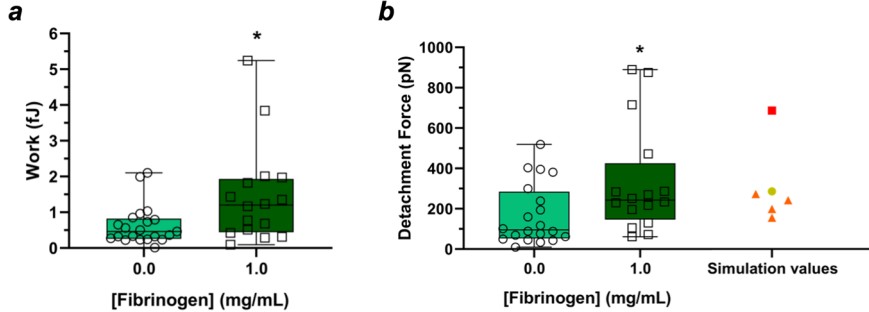

**Fig. 3 AFM data of erythrocyte–erythrocyte detachment work and maximum detachment force.** The work necessary to detach two erythrocytes in the presence of fibrinogen 1.0 mg/mL is higher than on its absence ($p = 0.030$) (**a**). The maximum detachment force of two erythrocytes in the presence of fibrinogen is also higher ($p = 0.030$) (**b**). For comparison, in **b** we also indicate the values of the erythrocyte–erythrocyte forces of our simulations. The yellow circle corresponds to the adhesion force between the cells in the simulation of Fig. 4 (below). The orange triangles indicate the adhesion forces between the cells in the simulations of Fig. 5 (below), for the different off-centred distances (1.02 μm, 2.04 μm, 3.06 μm and 4.08 μm). The red square depicts the adhesion force between the cells in the simulation of Fig. 6 (below). The experimental data refers to $n = 28$ biologically independent blood samples. For each blood sample approximately 40 force-distance curves were performed with and without fibrinogen addition. Experimental values are presented together with box-and-whiskers plots with the median value.

| Table 1 Model parameters. | | |
|---|---|---|
| **Parameter** | **Symbol** | **Value** |
| Bending rigidity | $K_B$ | 2.0 aJ |
| Bending rigidity coefficient | $\kappa_B$ | 432 J/m³ |
| Interface width (lattice size) | $\epsilon$ | 0.17 μm |
| Hard-core repulsion coefficient | $\gamma$ | 138 J/m³ |
| Area conservation coefficient | $\alpha_S$ | $1.8 \times 10^{-2}$ J/mm⁴ |
| Volume conservation coefficient | $\alpha_V$ | $5.9 \times 10^{4}$ J/mm⁶ |
| Target area | $S_{\text{target}}$ | 197 μm² |
| Target volume | $V_{\text{target}}$ | 160 μm³ |
| Mobility | $M$ | $7.4 \times 10^{-2}$ m³J⁻¹s⁻¹ |
| Velocity | $\|\vec{v}\|$ | 0.57 μm/s |

interaction between fibrinogen and its erythrocyte membrane receptor than on erythrocytes from healthy donors[17]. We also used AFM to show that higher work and force values are necessary to completely detach two erythrocytes from essential arterial hypertension patients, when compared with cells from healthy donors, both in the absence and presence of fibrinogen molecules in solution[19]. The erythrocytes from these hypertension patients were also stiffer and had a significant increase in aggregation, when compared with the control group[19].

The mathematical model permitted to visualise in detail the time dependence of the interaction forces and cell morphology alterations between two colliding erythrocytes. The adhesion forces and adhesion energies between the two cells have been quantitatively matched to the experimental assays. As the two erythrocytes approach each other, the adhesion works to maximise the contact area between the cells. The simulation enables then to follow the decrease in adhesion force as the cells are pulled away from each other. In situations of large adhesion, the alterations in morphology can be very significant. With the simulation, we can better understand, and explore in detail, the process of erythrocyte collision observed in the micropipette assay. Starting from the experimental mechanical characterisation of erythrocytes from a patient, we will be able to use this model to quantitatively predict the aggregation and flow dynamics of a group of those cells in a blood vessel.

The results obtained, in the previous works and here, high-lighting the changes on erythrocyte cell–cell interactions, may give important insights regarding health status, as well as the pathophysiological relevance of fibrinogen concentration and erythrocyte aggregation, since an increment on either of them may induce a reduction of the microcirculatory blood flow.

## Methods

**Donors.** Five healthy blood volunteers' donors were included in this study to perform the micropipette aspiration technique, following a protocol with Odense University Hospital (Denmark). This study was approved by the ethical committee of Odense University Hospital and by the University of Southern Denmark (SDU). Blood samples were obtained from the donors through the cooperation agreement on blood for research or quality assurance (Project number DP059, Application for blood for research or quality assurance, Document-620.8), at the South Danish Transfusion Service and Tissue Centre, Department of Clinical Immunology, Odense University Hospital. For the AFM studies, we used a control group of 28 healthy blood donors, following a protocol with Instituto Português do Sangue e da Transplantação (IPST, Lisbon, Portugal). The study was approved by the joint Ethics Committee of Faculdade de Medicina da Universidade de Lisboa and Centro Hospitalar Universitário Lisboa Norte. All the participants gave and signed their informed consent.

**Erythrocyte isolation and sample preparation.** Erythrocytes were isolated from the blood of healthy donors, as previously described[12,17,19]. The blood, collected into $K_3$EDTA anticoagulant tubes, was centrifuged at $1040 \times g$ for 10 min, at 22 °C. Plasma and buffy-coat were removed and the erythrocytes washed three times with buffered saline glucose citrate (BSGC) pH 7.3, supplemented with $CaCl_2$ 1 mM. Erythrocyte final suspension was prepared at 0.1% (v/v) haematocrit, in BSGC. For AFM experiments, the erythrocyte suspension was placed on a poly-L-lysine coated glass slide, and erythrocytes were allowed to deposit and adhere for 30 min, at room temperature. Non-adherent cells were removed by sequential washes with buffer. Cells were allowed to equilibrate for 15 min before the AFM measurements.

**Micropipette aspiration assay.** The micropipette aspiration system consisted of micron-sized glass pipettes operated by micromanipulators, a manometer system controlling the aspiration pressure, and a chamber on a microscope stage from which erythrocytes are aspirated into the micropipette[26,30,31]. The setup was built on an inverted optical microscope (Axiovert 100, Carl Zeiss, Jena, Germany) with 3D micrometres (Newport, Irvine, CA, USA) mounted on the stage that holds the micropipettes and the experimental chamber. Micropipettes were produced with single-barrel borosilicate capillary glass (0.75 mm × 0.4 mm 6″ glass capillary, A-M Systems, Sequim, WA, USA), heated and pulled using a pipette puller (P-47, Sutter Instruments, Novato, CA, USA). Tips were cut down to the desired inner size (1.5–2 μm) by using a micro forge (MF-900, Narishige, Setagaya, Japan)[51]. A chamber was placed on the slide-holder of the microscope for observation with an eyepiece between the camera and the microscope, using a 40 × magnification. The micropipettes were filled with BSGC supplemented with 1 mM $CaCl_2$ and connected to a water reservoir. Pressure control was monitored using a pressure transducer (Valydine Engineering, Northridge, CA, USA). After placing the chamber with buffer on the microscope and adjusting the micropipette, the size of its tip was recorded and the pressure adjusted according to the micrometre. The condition of no flow was considered as zero pressure. After introducing the erythrocyte suspension into the glass chamber, a small negative pressure (15 Pa) was applied, leading to the slow flow of an erythrocyte towards the pipette, enabling the cell to be caught at the pipette tip. The video was digitally recorded and operated with home-built LabVIEW programme. Videos were analysed with ImageJ software v. 1.53e (NIH, USA)[52].

The two-pipettes technique was applied to visualise the adhesion between two erythrocytes. We used this technique to observe the effects in erythrocyte shape as a consequence of erythrocyte–erythrocyte adhesion, both in the absence and

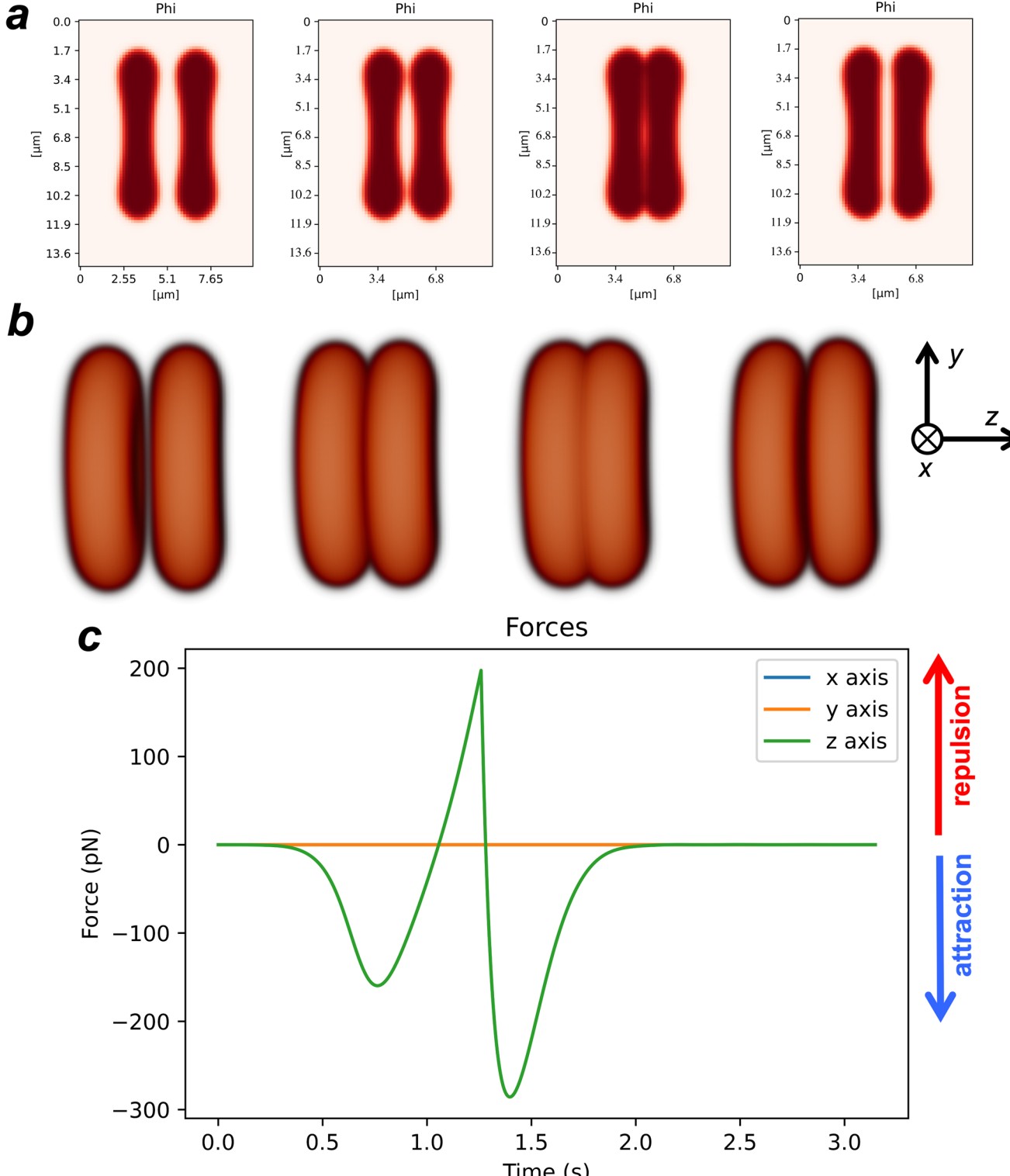

**Fig. 4 Simulation time-lapse of two aligned erythrocytes being pushed against each other and then pulled apart. a** Mid-slice profile, and **b** the corresponding 3D representation. The images correspond to the times $t = 0, 0.62, 1.25, 1.88$ s. **c** Total force applied on the right erythrocyte by the left erythrocyte as a function of time.

presence of fibrinogen (1 mg/mL). On all cell–cell adhesion experiments, a pressure of 20 Pa was maintained. A piezoelectrical driven linear translator (LISA, P-753.12C, Physik Instrumente, Germany, with modular piezo controller E500.00, Physik Instrumente, Germany) coupled to one of the micropipettes was used to control the movement of the erythrocytes (approach, adhesion and retraction). Using an arbitrary waveform generator (33220A, Agilent, Santa Clara, CA, USA), it was possible to programme the time course for piezo approach and retraction

cycles of the erythrocyte. Using a "ramp mode" with the time period of 15 s, velocity 1.14 μm/s, amplitude 3 V and symmetric regarding forward and backward motion, the observed deformation of the erythrocytes permitted to analyse the effects of cell–cell adhesion in the assay. These measurements were done in the absence and presence of 1.0 mg/ml fibrinogen, in BSGC with 1 mM of $CaCl_2$. Real-time video images were acquired and further analysed to determine the erythrocyte adhesion frequency.

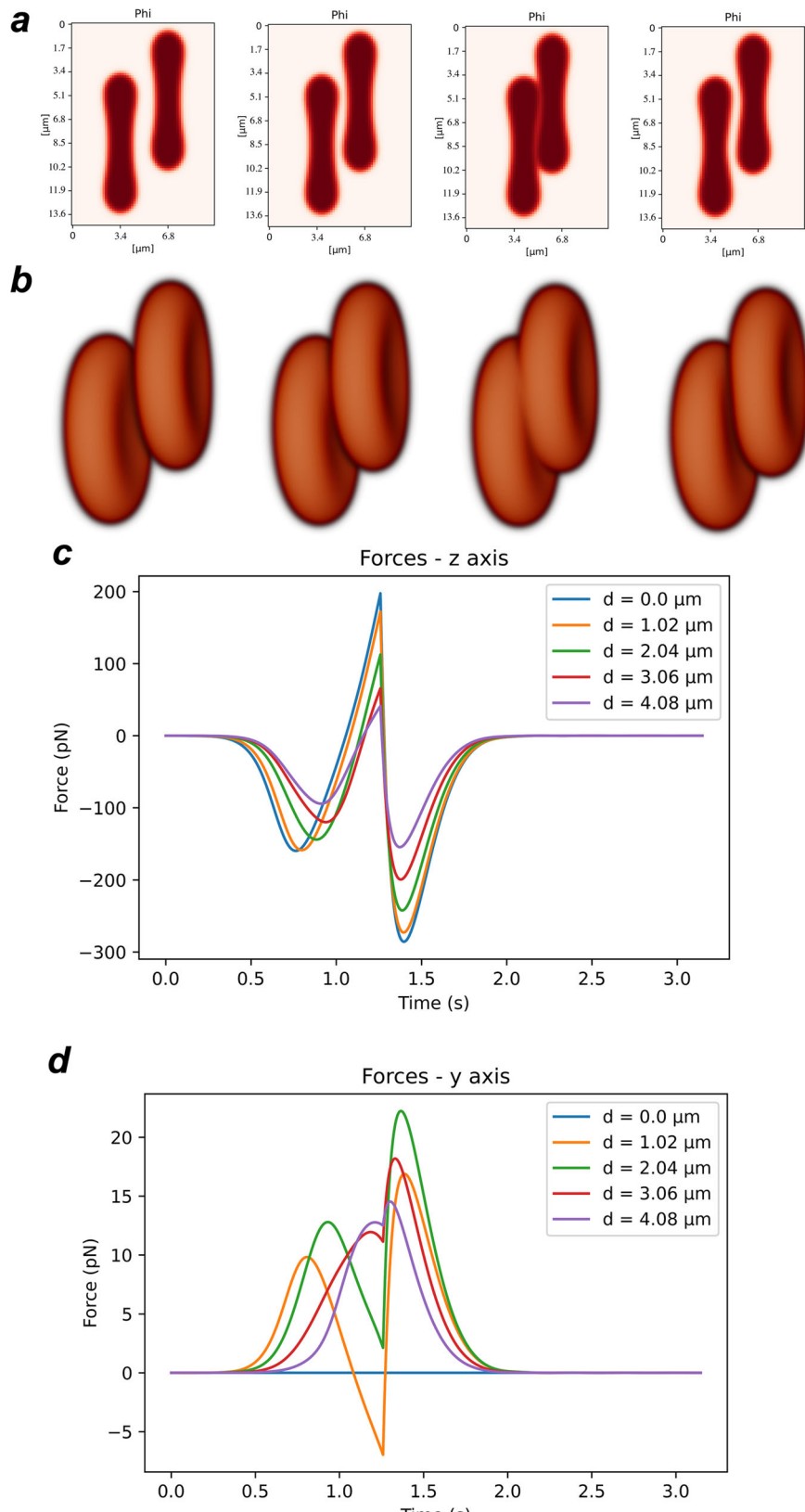

**Fig. 5 Simulation time-lapse of two erythrocytes being pushed against each other off centred and then pulled apart. a** Mid-slice profile and **b** corresponding 3D representation with offset distance $d = 3.06\,\mu m$. The images correspond to the times $t = 0, 0.62, 1.25, 1.88\,$s. Total force applied on the right erythrocyte by the left erythrocyte as a function of time (**c**) in the $z$ direction and **d** in the $y$ direction.

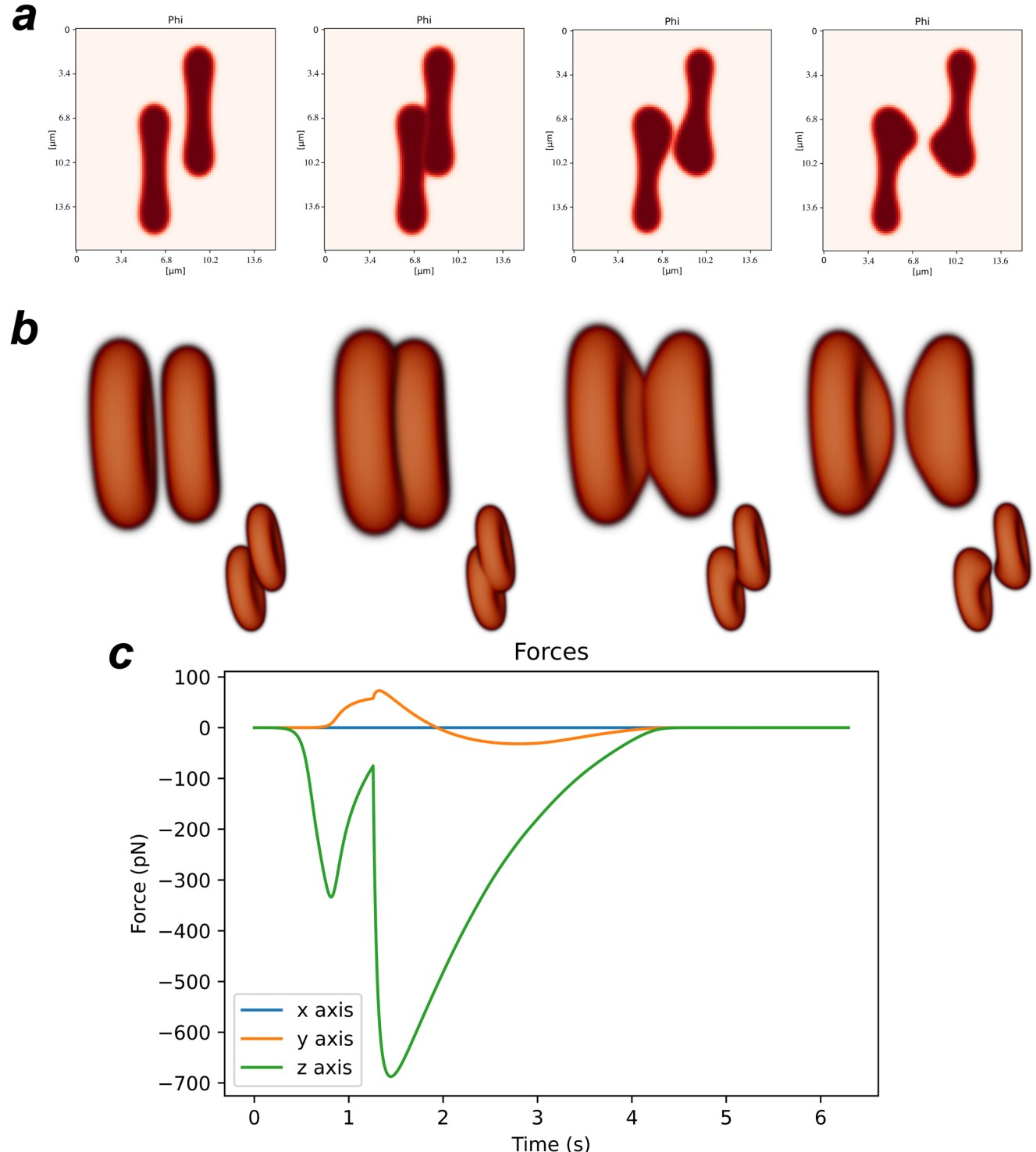

**Fig. 6 Simulation time-lapse of two offset erythrocytes being pushed against each other off centred and then pulled apart for large adhesion. a** Mid-slice profile and **b** corresponding 3D representation with offset distance $d = 4.08\,\mu m$ (at each time instance, we present the erythrocytes observed from two different perspectives: observed along the $y$ direction, in the larger images, and when observed along the $x$ direction, in the respective smaller images). The images correspond to the times $t = 0, 1.25, 3.13$, and $4.69$ s. **c** Total force applied on the right erythrocyte by the left erythrocyte as a function of the time.

**Atomic force microscopy.** AFM studies were performed on a JPK NanoWizard IV atomic force microscope with a CellHesion 200 module (JPK Instruments, Berlin, Germany), mounted on an Axiovert 200 inverted optical microscope (Zeiss, Jena, Germany). To conduct the AFM cell adhesion assays, tipless cantilevers Arrow TL2 (NanoSensors, Neuchatel, Switzerland), with 1 μm thickness, 0.03 N/m spring constant and 7 kHz resonance frequency were chemically functionalised, as previously described[18,22,53]. AFM cantilevers were cleaned with intense UV light for

15 min. Then, the cantilevers were incubated overnight in a 0.5 mg/mL biotinylated albumin solution, at 37 °C, in a humidified incubator. The cantilever was washed three times in phosphate buffered saline (PBS) pH 7.4 to remove unbound protein. After that, the cantilever was incubated in 0.5 mg/mL streptavidin (30 min, at room temperature) and, finally, in 0.4 mg/mL biotinylated-concanavalin A solution (30 min, at room temperature). Functionalised cantilevers were mounted on the microscope, the erythrocytes in suspension were injected, and one erythrocyte was

captured by positioning the cantilever above the cell centre and pressing onto the cell for ~30 s. The erythrocyte attached to the cantilever was then placed on top of other erythrocytes previously adhered to a poly-L-lysine glass slide, and the cell–cell adhesion experiments were performed. In this way, one erythrocyte was securely adhered to the cantilever, while the other was adhered to the surface, and in all force-distance curves of measurements on the erythrocyte–erythrocyte system (with and without the presence of fibrinogen) we never observed cell detachment from the tipless cantilever. In fact, according to[53], the strength of the concanavalin A attachment force is close to 2 nN, thus much stronger than the interaction force between two erythrocytes (below 1 nN, see Fig. 3b above). Moreover, poly-L-lysine is a positively charged amino acid residue polymer commonly adsorbed to surfaces for strong attachment to the negatively charged glycocalyx of cells, as erythrocytes and, it can provide a firm adherence of erythrocytes to the surface without changing their characteristic morphology [54].

The applied force was adjusted to 0.3 nN before retraction. Data collection for each force-distance cycle was performed at 2 μm/s, with a z-length of 30 μm. A retract and an extend delay of 5 s was applied. Measurements were performed in the presence or the absence of human fibrinogen (Sigma Aldrich, St. Louis, MO, USA). For each blood sample, approximately 40 force-distance curves were performed for each concentration (5 force-distance curves × 8 cells attached to a poly-L-lysine glass slides). Force-distance curves were analysed using JPK data processing software v. 6.055 (JPK Instruments, Berlin, Germany). Values of the maximum detachment force (by the minimum value of adhesion force) and of the work (energy) necessary to detach two erythrocytes were obtained after analysing each cell–cell adhesion retraction curves.

**Statistics and reproducibility**. Descriptive statistics are given as mean ± standard error of the mean (SEM). Results were analysed with Mann–Whitney $U$ test. $p < 0.05$ was considered as significant. Statistical analyses were performed using GraphPad Prism v. 8.02. The simulation results from Figs. 4–6 are the dirct result from the integration of Eq. (2).

**Mathematical model**. In the present mathematical model, we assign a scalar order parameter to each erythrocyte $\phi_i(\vec{r})$, where $i$ identifies the cell, such that $\phi_i \approx +1$ inside the respective erythrocyte and $\phi_i \approx -1$ outside. Near the cell membrane, $\phi_i(\vec{r})$ varies continuously, and $\phi_i = 0$ at the centre of the interface[33]. While this choice for equilibrium values of the scalar order parameter is standard in phase-field models that describe the bending rigidity of biological membranes and of other systems[50,55–57], the choice of $\phi_i \approx +1$ inside the cell and $\phi_i \approx 0$ outside may be found in other works in the field[42,44,45].

The most relevant energies involved in determining erythrocytes' shapes and dynamics are the bending energy due to curvature, the adhesion energy when the erythrocytes approach, and the hard core repulsion to prevent the inter-penetration between two erythrocytes. The cell rigidity provided by the erythrocyte cytoskeleton provides further stability to the erythrocyte shape, reducing membrane fluctuations and influencing the measured values of the bending rigidity and membrane tension[58]. As in this work the focus is in exploring the time dependence of the cell–cell interaction forces at the surface of the erythrocytes, we will focus the model on the description of cell adhesion and hard core repulsion. We observe that the bending rigidity energy term is sufficient to stabilise the oblate erythrocyte characteristic shape[50,59,60] without needing the cytoskeleton mechanics. Hence, we will consider in the model the bending rigidity $K_B = 2.0$ aJ, which is inside the typical range of erythrocytes bending rigidities, from 0.2 to 3 aJ[61–65], but for the sake of simplicity, we will not add an explicit description of the cell's rigidity provided by the cytoskeleton.

We initially implement a free energy functional $F[\phi_1(\vec{r}), \phi_2(\vec{r})]$ that takes into account the considered energetic contributions [42,44,46,50,59] (derived from the Cahnam-Helfrish bending energy[66]):

$$F[\phi_1(\vec{r}), \phi_2(\vec{r})] = \underbrace{\int \kappa_B \sum_{i=1}^{2} \left(-\phi_i + \phi_i^3 + \epsilon^2 \nabla^2 \phi_i\right)^2 \mathrm{d}\vec{r}}_{F_{\text{bending}}}$$
$$+ \underbrace{\int \left(\gamma h(\phi_1) h(\phi_2) + \eta \nabla h(\phi_1) \cdot \nabla h(\phi_2)\right) \mathrm{d}\vec{r}}_{F_{\text{interaction}}}$$
$$+ \underbrace{\alpha_S \sum_{i=1}^{2} \left(S_{\text{target}} - S[\phi_i]\right)^2}_{F_{\text{Surface}}}$$
$$+ \underbrace{\alpha_V \sum_{i=1}^{2} \left(V_{\text{target}} - V[\phi_i]\right)^2}_{F_{\text{volume}}}, \quad (1)$$

where $\kappa_B$, $\gamma$ and $\eta$ are coefficients proportional to the bending rigidity of the cell membrane[50], to the hard-core repulsion between cells and to the cell–cell surface adhesion[42,46,50], respectively. The width of the interface in the model is given by the coefficient $\epsilon$[33], which also relates $\kappa_B$ to the membrane bending rigidity in the sharp interface limit, $K_B = \frac{4\epsilon^3}{3\sqrt{2}} \kappa_B$[50].

The first term of the free energy describes the bending energy of the erythrocyte and, in the sharp interface limit, becomes identical to the mean curvature term of the Cahnam-Helfrish energy, i.e., to $\int ds K_B C^2$, where the integral is done over the membrane and $C$ is the membrane local mean curvature. The Gaussian curvature term of the Cahnam-Helfrish energy is not included in our phase-field description of erythrocyte dynamics, since these cells conserve their integrity in our experiments without changing their topology, and since, due to the Gauss-Bonnet theorem, the integral of the Gaussian curvature in a closed vesicle is a topological invariant[67]. Therefore, this term would only affect the cell dynamics if the Gaussian curvature rigidity coefficient was not constant throughout the cell membrane, or if topological transformations were present during the dynamics. To obtain the $F_{\text{bending}}$ term, the local curvature at a specific point of the phase field interface is written as a function of the order parameter and its derivatives, and then integrated over the system volume (see ref. [50] for a detailed derivation).

The hard-core repulsion term between the two erythrocytes leads to an increase in the system's energy when the erythrocytes overlap in space. In this term, the hard-core repulsion coefficient $\gamma$ will be set to a value large enough to prevent the overlap of the cells. The function $h(\phi_i) = \frac{1}{4}(1 + \phi_i)^2(2 - \phi_i)$ is approximately equal to 1/2 at the vicinity of $\phi_i = 0$, and has extrema at $h(-1) = 0$ and $h(1) = 1$. This function moderates the hard-core repulsion term, which would otherwise be responsible for pushing the order parameter values away from the interval $[-1, 1]$[42].

The second term of the interaction part of the system's free energy (1) describes the surface adhesion between the erythrocytes. This term indicates that the energy decreases when the two order parameter gradients are anti-parallel, i.e., when the cells are in contact.

Finally, the surface and volume terms penalise the system energetically when the erythrocyte surface and volume is different from their target values $S_{\text{target}}$ and $V_{\text{target}}$. The surface and volume of each erythrocyte can be directly obtained from the following functionals, $S[\phi_i] = \frac{3}{2\sqrt{2}} \epsilon \int \left(\nabla \phi(\vec{r})\right)^2 \mathrm{d}\vec{r}$ and $V[\phi_i] = \int h(\phi) \mathrm{d}\vec{r}$, respectively. The parameters $\alpha_S$ and $\alpha_V$ in Eq. (1) are penalty parameters, which will be set large enough that both the volume and surface of the erythrocytes are kept constant during the system's evolution (see Eq. (2) below).

From this free energy, we can obtain the following equation for the time evolution of the order parameter $\phi_1$ (the equation for the evolution of the other erythrocyte is obtained from (2) by exchanging the indices 1 and 2):

$$\frac{\partial \phi_1}{\partial t} + \vec{v} \cdot \nabla \phi_1 = -M \frac{\delta F}{\delta \phi_1} = -M\Big[2\left(3\phi_1^2 - 1\right)\kappa_B\left(-\phi_1 + \phi_1^3 - \epsilon^2 \nabla^2 \phi_1\right)$$
$$- 2\kappa_B \epsilon^2 \nabla^2\left(-\phi_1 + \phi_1^3 - \epsilon^2 \nabla^2 \phi_1\right)$$
$$+ \frac{3}{4}\gamma\left(1 - \phi_1^2\right) h(\phi_2) - \frac{3}{4}\eta\left(1 - \phi_1^2\right)\nabla^2 h(\phi_2)\Big]$$
$$- 3\sqrt{2} M \epsilon \alpha_S\left(S_{\text{target}} - S[\phi_1]\right)\nabla^2 \phi_1$$
$$+ \frac{3}{2} M \alpha_V\left(V_{\text{target}} - V[\phi_1]\right)\left(1 - \phi_1^2\right), \quad (2)$$

where $\vec{v}$ is the advection velocity of the cell and $M$ is the system's mobility. A dimensionless version of Eq. (2) can be written by noting that $\tau = (M\kappa_B)^{-1}$ defines the simulation timescale, and the interface width, $\epsilon$, gives the simulation length scale. The time and space variables in Eq. (2) can be then expressed as $t = t'\tau$, $x = x'\epsilon$, $y = y'\epsilon$, and $z = z'\epsilon$, where $t'$, $x'$, $y'$, and $z'$ are dimensionless variables. We finally observe that the relevant parameters that describe the system dynamics are the dimensionless advection velocity, $v' = \frac{v}{M\kappa_B\epsilon}$, the dimensionless repulsion coefficient, $\gamma' = \frac{\gamma}{\kappa_B}$, and the dimensionless adhesion coefficient, $\eta' = \frac{\eta}{\kappa_B\epsilon^2}$. Nevertheless, in this work we will be using Eq. (2) in its dimensional form for a more direct substitution of the experimental parameters.

The parameters for the model are indicated in Table 1.

Finally, the interaction force in one of the cells due to the other erythrocyte, given by ref. [33]:

$$\vec{F_1} = -\int \phi_1 \nabla \frac{\delta F_{\text{interaction}}}{\delta \phi_1} \mathrm{d}\vec{r} =$$
$$= -\int \frac{3}{4} \phi_1 \nabla\left(\gamma\left(1 - \phi_1^2\right) h(\phi_2) - \eta\left(1 - \phi_1^2\right)\nabla^2 h(\phi_2)\right)\mathrm{d}\vec{r}. \quad (3)$$

**Numerical methods**. Initially, the shape of each isolated cell was obtained from the integration of Eq. (2) by setting the velocity and cell–cell interaction terms to zero. Then, with the full equation, the collision between the two cells was simulated in a 3D box of size $15.3 \times 15.3 \times 15.3$ μm³. We have used a fixed grid spectral semi-implicit method to integrate Eq. (2). The non-linear terms were initially calculated in real space, and the fast-Fourier transform algorithm was used to obtain the 3D Fourier transform of these terms. The next iteration values for the order parameters $\phi_1$ and $\phi_2$ were calculated directly in Fourier space. This was done semi-implicitly with the linear time derivative and fourth spatial derivative (squared Laplacian) terms being written as a function of the order parameters in the next time step, and the other terms being written as a function of the order parameters in the present time step.

**Reporting summary**. Further information on research design is available in the Nature Portfolio Reporting Summary linked to this article.

## Data availability

The data that is plotted in Figs. 2 and 3 of this study are available in the Supplementary Data file.

## Code availability

The results presented in Figs. 4–6 are the representation of the direct integration of equation (2). The home-built code used to integrate these equations is available in the Supplementary Data file.

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

## Acknowledgements

This work was supported by Fundação para a Ciência e a Tecnologia – Ministério da Ciência, Tecnologia e Ensino Superior (FCT-MCTES, Portugal) projects PTDC/EMD-TLM/7289/2020 (all authors), PTDC/BBB-BMD/6307/2014 and UID/BIM/50005/2019 (CSL, FAC, NCS), through Fundos do Orçamento de Estado. JC and RT also thank the support of FEDER funds through the Operational Programme Competitiveness Factors – COMPETE and Fundação para a Ciência e a Tecnologia through the strategic projects UIDB/04564/2020 and UIDP/04564/2020. CSL also acknowledges FCT-MCTES fellowships PD/BD/135045/2017 and COVID/BD/151823/2021, General Programme of the European Molecular Biology Conference – EMBO Short-Term Fellowship 8350, and an EBSA Bursary for a working visit to the University of Southern Denmark to perform the micropipette aspiration technique experiments, as well as Erasmus+ mobility – ERASMUS SMT Student Mobility for Traineeships. AH-M acknowledges support from Ministerio de Ciencia e Innovación (Spain) under project PID2019-106063GB-100 and AGAUR (Generalitat de Catalunya) under project 2017 SGR-1061.

## Author contributions

R.D.M.T. and N.C.S. designed and supervised the experiments, modelling and data analyses. K.K. designed and developed the adhesion measurement with the micropipette manipulation technique. F.A.C. optimised, performed and supervised the AFM cell–cell adhesion experiments. A.H.M. and R.D.M.T. developed the mathematical model. C.S.L. performed the micropipette aspiration and AFM cell–cell experiments. J.C. wrote the code, ran the simulations and represented the simulation results. All authors analysed the data, discussed the results, and wrote the manuscript.

## Competing interests

The authors declare no competing interests.
