## [Peer Review File · Communications Biology]

Reviewers' comments:

Reviewer #1 (Remarks to the Author):

This is a well-written manuscript with studies of RBC cell-cell interactions and the effect of fibrinogen on this interaction. The authors used two state-of-the-art methods, an AFM force-pulling method that they previously described and a new micropipette aspiration method to manipulate individual cells and make them interact. This is complemented by a molecular modelling method to analyse different types of interaction. Consistently the authors find that fibrinogen enhances RBC-RBC interaction in both experimental methods and the experimental methods feed relevant parameters into the molecular mathematical model. This is an elegant study from an experienced group studying these interactions. I have the following suggestions and comment, some of which relate to the relative comparison of forces and their implications for biology.

- In AFM studies, Tips were functionalised with concavalin a and used to capture RBC. The other RBC was immobilised on a lysine surface. How strong are the relative concavalin a and lysine surface binding forces to RBCs respectively compared with the fibrinogen-RBC interaction? How do we know that the rupture analysed involves RBC-fibrinogen-RBC detachment and not detachment of RBC from the functionalised AFM tip or from the lysine surface?

- A figure depicting the experimental set-up would be useful, for both the AFM and micropipette RBC-fibrinogen-RBC interaction experiments.

- The panel C of fig 3 is called for first in the main text. Figure panels should be cited sequentially. Same for Fig 4C.

- Normal RBC rouleaux formation that occurs in static blood as discussed in the background seems to favour aligned cell-cell interaction as depicted in Fig 3 rather than the "staggered" interactions of Fig 4 and 5. Do the force analysis measurements agree with a favoured aligned cell-cell interaction as observed in RBC rouleaux?

Reviewer #2 (Remarks to the Author):

This paper uses a phase-field model of RBC dynamics that allows to study adhesion and impact of multiple RBCs. The model is calibrated using experiments. Once the model has been calibrated, the authors use it understand details of the adhesion and separation of previously adhered RBCs. The paper is well written. Overall, I found this manuscript interesting, and adequate for Communications Biology. I have some comments to improve what is already a good manuscript:

1. The authors discuss qualitatively how their numerical results compare with the experiments in Fig. 2. I think that instead of that approximate comparison it would be much more informative to compute the integral of the pointwise force from the simulations and plot the results also in Fig. 2, next to the experimental results.

2. In the model description, the authors define α_s and α_v as Lagrange multipliers, but later they say that (rather than computing their values) they simply take very large values of the alphas to make sure the constraint is satisfied. I understand why they may want to do that, but in that case the alphas are not Lagrange multipliers, they are simply penalty parameters.

Reviewer #3 (Remarks to the Author):

This paper is about experiments and mathematical research on red blood cells. Experimental studies quantitatively measure cell-to-cell adhesion in the presence (or absence) of fibrinogen using a micropipette. In this experiment, the author obtained some new measurement results such as the elastic modulus of the membrane and the viscosity of the membrane, and I support all

these results. The author approaches the morphology of red blood cells not only from experimental studies but also from mathematical aspects. In general, the method using a triangular mesh is relatively often used for representing the shape of red blood cells, but in this paper, the author constructs a new method using the Phase Field method. This has the potential to become a standard method for investigating various problems of morphogenesis. In that sense, I believe that this paper is a good example of cross-disciplinary and integrated research between mathematical science and biology.

0: Overall, I have few objections. The article is generally well organized, and I don't think it will be difficult for readers with a high level of expertise to understand the content. On the other hand, for general readers of this journal, I thought it would be helpful to explain with some more examples.

1: The order variable ϕ is often used in cell shape studies using the Phase Field method. At this time, $\phi=1$ is often used in areas where cells exist, and $\phi=0$ is used in areas where cells do not exist. This is the same for crystal growth models using the Phase Field method. In this paper, $\phi=-1$ is used in nonexistent regions. This can be reduced to the $\phi=0$ case with appropriate variable transformations, so the form of the expression does not matter in either case. However, since the formula is slightly different from the formula used in the references, the reader may be a little confused. Is it possible to replace the variables in the paper?

2: The author cites Nonomura (2012), after which a new method using the Phase Field method was developed by the same authors (<https://iopscience.iop.org/article/10.1088/1478-3975/aaee45>). Please consider citing this paper as there are claims that overlap with this paper.

3: I think Cahn-Helfrich bending energy is a model including mean curvature and Gaussian curvature. On the other hand, the first term on the right-hand side of Eq. (1) expresses the bending energy using the variable $\phi_{i,j}$, which, at least for me, is a leap from the Cahn-Helfrich form of bending energy. I finally understand that the bending energies of this paper and the Cahn-Helfrich bending energies are qualitatively the same. However, for the benefit of the general reader, please specify the process of formula derivation.

3-1: Also, is $\nabla^2\phi_{i,j}$ in expression (1) a typo of $|\phi_{i,j}|^2$?

4: Since $\phi_{i,j}$ is an order variable, its value should be between -1 and 1. On the other hand, from the author's energy form of F in equation (1), it is expected to have local minima at -1 and 1. However, by changing the parameters, it is possible that the energy functional does not have local minima at -1 and 1. If any parameter has local minima at -1 and 1, mention this. Otherwise, please ensure that the parameters used in this paper have a local minimum at -1 and 1.

5: For the model to be robust, I think the author should examine the parameter sensitivity of the model. For example, Bending rigidity is 2.0, will changing this a little have a big impact on the results? Or are the results the same? I would like other parameters to be investigated as well. Please mention this.

6: Although it is somewhat related to 5, I would like to know the mechanism that plays an essential role in this phenomenon through mathematical formulas. For this reason, nondimensionalization of expressions is an important process. Show the dimensionless equation. This process also allows the reader to know the essential parameters.

7: The numerical calculation method seems to be almost appropriate. On the other hand, I didn't quite understand how to handle the advection term in the second term on the left side of equation (2). Advection terms can easily lead to numerical instabilities, so the author used some method (upwind differencing scheme??), didn't he? Please mention this.

Response COMMSBIO-22-2900-T

We thank the three reviewers for their comments and questions and in this document, we proceed to reply carefully to the different points raised by them.

Reviewer #1

1. In AFM studies, Tips were functionalised with concavalin a and used to capture RBC. The other RBC was immobilised on a lysine surface. How strong are the relative concavalin a and lysine surface binding forces to RBCs respectively compared with the fibrinogen-RBC interaction? How do we know that the rupture analysed involves RBC-fibrinogen-RBC detachment and not detachment of RBC from the functionalised AFM tip or from the lysine surface?

AFM-based force spectroscopy is a method particularly appropriate to determine the interaction forces between fibrinogen and human blood cells because the process of cell isolation is not an issue with this technique, as the measurements are conducted at the single-cell level. After isolating the erythrocytes, the cell-cell interactions are measured with an erythrocyte attached to a cantilever positioned on the top of a single erythrocyte at a time (see new Fig. 1), while it can be optically imaged in real-time. This procedure assures that the blood cells are not removed from the cantilever and/or from the surface. The positioning of the cantilever on top of a single cell was done with the help of a CCD camera. This camera enables the imaging of the cells under evaluation, reassuring us that their detachment from the AFM cantilever or from the solid surface does not occur.

From Figure 2B (now Figure 3B) of the manuscript, it is possible to observe that the force of adhesion between two erythrocytes in the presence of fibrinogen is up to 500 pN, increasing to up to 900 pN in the presence of fibrinogen 1 mg/mL. Wojcikiewicz *et al.* (Wojcikiewicz EP *et al.* Biol Proced Online 6 (200): 1-9) reported that the strength of the concanavalin A attachment force is close to 2 nN, thus much stronger than the interaction force between two erythrocytes. This is crucial for this type of measurements. Otherwise, the erythrocyte could come off the tip during cell-cell interaction measurements. In fact, we were able to obtain 40 force-distance curves of measurements on the erythrocyte-erythrocyte system (with and without the presence of fibrinogen) without cell detachment from the tipless cantilever.

Cells were also firmly attached to poly-L-lysine coated glass slide after 30 min of deposition, at room temperature. Non-adherent cells were removed by sequential washes with buffer, loaded into the AFM and allowed to equilibrate in the buffer for 15 min before force spectroscopy measurements. Poly-L-lysine is a positively charged amino acid residue polymer commonly adsorbed to surfaces for strong attachment to the negatively-charged glycocalyx of cells, as erythrocytes. At a typical 1 mg/ml concentrations of poly-L-lysine pre-adsorbed to a glass slide, as the ones used in this study, erythrocytes adhere firmly and retain their characteristic morphology (Hategan A *et al.* Biophys J 85 (2003): 2746-59).

This information is now included in the main text of the manuscript, on page 6.

“In this way, one erythrocyte was securely adhered to the cantilever, while the other was adhered to the surface, and in all force-distance curves of measurements on the erythrocyte-erythrocyte system (with and without the presence of fibrinogen) we never observed cell detachment from the tipless cantilever. In fact, according to [51], the strength of the concanavalin A attachment force is close to 2 nN, thus much stronger than the interaction force between two erythrocytes (below 1 nN, see Fig. 3B below). Moreover, poly-L-lysine is a positively charged amino acid residue polymer commonly adsorbed to surfaces for strong attachment to the negatively-charged glycocalyx of cells, as erythrocytes and, it can provide a firm adherence of erythrocytes to the surface without changing their characteristic morphology [52].”

2. A figure depicting the experimental set-up would be useful, for both the AFM and micropipette RBC-fibrinogen-RBC interaction experiments.

We agree with the Reviewer suggestion. For that, we now add a new Figure 1 to the manuscript with the schematic experimental setup representation of the erythrocyte-erythrocyte adhesion experiments with micropipette aspiration technique and atomic force microscopy.

A - Micropipette Aspiration

B - Atomic Force Microscopy

“Fig. 1 Schematic experimental setup representation of the erythrocyte-erythrocyte adhesion experiments carried out using micropipette aspiration (A) and atomic force microscopy (B). Using the micropipette aspiration technique (A), a negative pressure (suction) is applied to catch an erythrocyte in each micropipette. One of these micropipettes (depicted as the one on the left side of each panel from a to d) is kept static. The piezo-driven micropipette, with an attached erythrocyte, approaches the other erythrocyte on the static micropipette, until a certain contact force is reached (a). Then, during the retraction of the right side pipette, cell-cell adhesion may occur, leading to the pulling of the surface of the erythrocyte from the left side when the erythrocyte from right side micropipette is being moved away from it (b and c), until the complete erythrocyte-erythrocyte detachment is reached (d). Using atomic force microscopy (AFM) (B), the tipless cantilever with an attached erythrocyte is approached to other erythrocyte on the substrate, until a certain contact force is reached (a). Afterwards, during retraction, cell-cell adhesion may occur, causing the cantilever to adhere to the sample up to a distance beyond the initial contact point (b). When the applied force overcomes the cell-cell interaction forces, the cantilever pulls off sharply (c), before totally breaking the cell-cell contact and complete the erythrocyte detachment (d).”

3. The panel C of fig 3 is called for first in the main text. Figure panels should be cited sequentially. Same for Fig 4C.

The description of the now called Fig. 4 starts only on the second paragraph of page 14. There, the panels are described in order: 4A, 4B and 4C. Before the description of Fig. 4, we justify the choice for the attraction coefficient, that was set to $\eta = 8.0 \times 10^{-12}$ J/m to simulate attractive forces in the range of 300 pN, as observed in the AFM assays. Here, we should mention Fig. 3B, but we also need to refer that below, in the simulations results, we obtain an adhesion force between the two cells within this range of magnitude. We now mention both Fig. 3B and Fig. 4C.

The plates in the now called Fig. 5 are described in order. On paragraph 2 of page 15, we describe Fig. 5A and B (however, we rewrote that part of the text to be more explicit) and Fig. 5C is described only afterwards, on paragraph 3 of page 15.

4. Normal RBC rouleaux formation that occurs in static blood as discussed in the background seems to favour aligned cell-cell interaction as depicted in Fig 3 rather than the “staggered” interactions of Fig 4 and 5. Do the force analysis measurements agree with a favoured aligned cell-cell interaction as observed in RBC rouleaux?

Both AFM-based force spectroscopy and the micropipette aspiration techniques are single-cell techniques. They only assess the interactions occurring between two erythrocytes, which could be extrapolated to the formation of a normal erythrocyte rouleaux. Erythrocyte rouleaux formation refers to the stacking of 4 or more erythrocytes. Erythrocytes membranes have a negative surface potential charge that causes the cells to repel each other. In the presence of increased concentrations of proteins able to bridge or connect erythrocyte (e.g., fibrinogen), the ability of erythrocytes to stick together is promoted. The alignment of the cell-cell interaction depicted in Fig. 4 is the perfect alignment of the erythrocyte in the rouleaux. However, sometimes the cells alignment may be deviated from the center of both erythrocytes and the erythrocyte rouleaux formation will result from cell-cell contacts as the ones simulated in Figures 5 and 6. Svetina et al. (Svetina S et al. Bioelectrochemistry 73 (2008): 84-91) defined different erythrocyte-erythrocyte adhesion regimes. They explained that on a weak adhesion regime, the erythrocytes could be aligned both in a zig-zag or staircase cell sequence and on a strong-adhesion

regime the erythrocyte rouleaux growth is linear, which is the usual preferred mode of rouleaux formation. Other authors have already reported examples of different erythrocytes rouleaux alignments (Zehe A et al. Braz J Med Biol Res. 37 (2004): 173-83; Steffen P et al. Phys Rev Lett. 110 (2013): 018102; Wagner C et al. Comptes Rendus Physique 14 (2013): 459-469; Flormann D et al. Sci Rep. 7 (2017): 7928).

Reviewer #2

1. The authors discuss qualitatively how their numerical results compare with the experiments in Fig. 2. I think that instead of that approximate comparison it would be much more informative to compute the integral of the pointwise force from the simulations and plot the results also in Fig. 2, next to the experimental results.

We have now included in Fig 3B the attachment forces simulated for comparison.

The new Figure caption reads:

“Fig. 3 AFM data of erythrocyte-erythrocyte detachment work (A) and maximum detachment force (B). The work necessary to detach two erythrocytes in the presence of fibrinogen 1.0 mg/mL is higher than on its absence ($p = 0.030$) (A). The maximum detachment force of two erythrocytes in the presence of fibrinogen is also higher ($p = 0.030$) (B). Values are presented as box-and-whiskers plot with the median value. For comparison, in plate (B) we indicate the values of the erythrocyte-erythrocyte forces of our simulations. The yellow circle corresponds to the adhesion force between the cells in the simulation of Fig. 4 (below). The orange triangles indicate the adhesion forces between the cells in the simulations of Fig. 5 (below), for the different off-centered distances (1.02 μm , 2.04 μm , 3.06 μm , and 4.08 μm). The red square depicts the adhesion force between the cells in the simulation of Fig. 6 (below).”

2. In the model description, the authors define α_s and α_v as Lagrange multipliers, but later they say that (rather than computing their values) they simply take very large values of the alphas to make sure the constraint is satisfied. I understand why they may want to do that, but in that case the alphas are not Lagrange multipliers, they are simply penalty parameters.

We thank the Reviewer for pointing this in correction. The coefficients α_S and α_V are indeed penalty parameters. We have altered the text accordingly.

Reviewer #3

1: The order variable ϕ is often used in cell shape studies using the Phase Field method. At this time, $\phi=1$ is often used in areas where cells exist, and $\phi=0$ is used in areas where cells do not exist. This is the same for crystal growth models using the Phase Field method. In this paper, $\phi=-1$ is used in nonexistent regions. This can be reduced to the $\phi=0$ case with appropriate variable transformations, so the form of the expression does not matter in either case. However, since the formula is slightly different from the formula used in the references, the reader may be a little confused. Is it possible to replace the variables in the paper?

The Reviewer is right that, in many works using the phase-field model to simulate cell dynamics, $\phi = 0$ is commonly associated to the regions outside the cell. However, most of those works simulate the cell membrane as an interface with a surface tension, and not as a membrane with a bending rigidity. With only few exceptions (the works of Herbert Levine, for

example), applications of phase-field in systems where the interfaces have a bending rigidity use $\phi = -1$ in the regions outside the domain (see for example Gu et al. *Journal of Mathematical Biology* 73 (2016): 1293-1319; Marth, et al. *Journal of Mathematical Biology* 69 (2014): 91-112; Valizadeh, et al. *Computer Methods in Applied Mechanics and Engineering* 388 (2022): 114191; Lázaro, et al. *Chemistry and Physics of Lipids* 185 (2015): 46-60). This is the standard choice in the community both in applications to cell dynamics and in other mechanical engineering systems.

The phase field description of the bending rigidity is rather complex mathematically, and the choice of $\phi = \pm 1$ in the equilibrium states simplifies the equations. We think that a strong point of our manuscript is the explicit description of the model used (equation (2)). The choice of $\phi = 0, 1$ would unfortunately both make the equation longer and more foreign to people working with bending rigidity in phase-field modeling.

To remind the reader that a different choice for the order parameter equilibrium values can be made, we have added the following sentence to page 7:

“While this choice for equilibrium values of the scalar order parameter is standard in phase-field models that describe the bending rigidity of biological membranes and of other systems [53–56], the choice of $\phi_i \approx +1$ inside the cell and $\phi_i \approx 0$ outside may be found in other works in the field [42, 44, 45].”

2: The author cites Nonomura (2012), after which a new method using the Phase Field method was developed by the same authors (<https://iopscience.iop.org/article/10.1088/1478-3975/aaee45>). Please consider citing this paper as there are claims that overlap with this paper.

We now cite the Akiyama et al. (2018) as well. We thank the Reviewer for pointing out this interesting and relevant application of the initial model.

3: I think Cahn-Helfrich bending energy is a model including mean curvature and Gaussian curvature.

Correct. Notice that the Gaussian curvature term is not included in the phase-field description since, due to the Gauss-Bonnet theorem, the integral of the Gaussian curvature in a closed vesicle is a constant (see Campelo et al. *The European Physical Journal E* 20 (2006): 37–45, and Rueda-Contreras et al. *Scientific Reports* 11 (2021): 1-10). Therefore, this term does not affect the cell dynamics (unless the Gaussian curvature rigidity coefficient is not constant throughout the cell membrane).

On the other hand, the first term on the right-hand side of Eq. (1) expresses the bending energy using the variable ϕ_i , which, at least for me, is a leap from the Cahn-Helfrich form of bending energy. I finally understand that the bending energies of this paper and the Cahn-Helfrich bending energies are qualitatively the same. However, for the benefit of the general reader, please specify the process of formula derivation.

We thank the Reviewer for raising this point. The derivation of equation (1) is rather involved, and it is detailed in Campelo et al. *The European Physical Journal E* 20, 37–45 (2006). However, we have now resumed the main steps in page 8.

“The first term of the free energy describes the bending energy of the erythrocyte and, in the sharp interface limit, becomes identical to the mean curvature term of the Cahn-Helfrich energy, i.e. to $\int ds K_B C^2$, where the integral is done over the membrane and C is the membrane local mean curvature. The Gaussian curvature term of the Cahn-Helfrich energy is not included in our phase-field description of erythrocyte dynamics since these cells conserve their integrity in our experiments without changing their topology, and since, due to the Gauss-Bonnet theorem, the integral of the Gaussian curvature in a closed vesicle is a topological invariant [66]. Therefore, this term would only affect the cell dynamics if the Gaussian curvature rigidity coefficient was not constant throughout the cell membrane, or if topological transformations were present during the dynamics. To obtain the F_{bending} term, the local curvature at a specific point of the phase field interface is written as a function of the order parameter and its derivatives, and then integrated over the system volume (see [53] for a detailed derivation).”

3-1: Also, is $\nabla^2 \phi_i$ in expression (1) a typo of $|\phi_i|^2$?

In the phase-field model describing bending rigidity, the coefficient in the free energy is $\kappa_B \nabla^2 \phi$. The manuscript is correct.

4: Since ϕ_i is an order variable, its value should be between -1 and 1. On the other hand, from the author's energy form of F in equation (1), it is expected to have local minima at -1 and 1. However, by changing the parameters, it is possible that the energy functional does not have local minima at -1 and 1. If any parameter has local minima at -1 and 1, mention this. Otherwise, please ensure that the parameters used in this paper have a local minimum at -1 and 1.

We thank the Reviewer for this comment. The parameter values simulated do not significantly shift the order parameter inside and outside the cells from the equilibrium values $\phi = \pm 1$. In fact, we only observe small differences well below 1% (in the third decimal place).

5: For the model to be robust, I think the author should examine the parameter sensitivity of the model. For example, Bending rigidity is 2.0, will changing this a little have a big impact on the results? Or are the results the same? I would like other parameters to be investigated as well. Please mention this.

6: Although it is somewhat related to 5, I would like to know the mechanism that plays an essential role in this phenomenon through mathematical formulas. For this reason, nondimensionalization of expressions is an important process. Show the dimensionless equation. This process also allows the reader to know the essential parameters.

We thank the Reviewer for these interesting questions, which we will answer together. The whole equation may be divided by $M\kappa_B = 1/\tau$, which provides the time scale of the dynamics. The relevant parameters that describe the dynamics are therefore the dimensionless velocity, $v' = \frac{v}{M\kappa_B\epsilon}$, the dimensionless repulsion coefficient, $\gamma' = \frac{\gamma}{\kappa_B}$, and the dimensionless attraction coefficient, $\eta' = \frac{\eta}{\kappa_B\epsilon^2}$, with ϵ being the length scale associated to the interface width. The magnitude of the repulsive and attractive forces per unit area (from equation (3)) will be proportional to $\gamma'\kappa_B = \gamma$ and to $\eta'\kappa_B = \frac{\eta}{\epsilon^2}$, respectively. Therefore, the forces between the cells will be mainly dependent on the coefficients γ and η . Alterations in the bending rigidity determine membrane deformation and its relaxation velocity after deformation, and thus will be much less determinant for the force magnitude between the erythrocytes than γ and η . We now provide information regarding the dimensionless parameters in the text on the bottom of page 9:

“A dimensionless version of equation (2) can be written by noting that $\tau = (M\kappa_B)^{-1}$ defines the simulation timescale, and the interface width, ϵ , gives the simulation length scale. The time and space variables in equation (2) can be then expressed as $t = t'\tau$, $x = x'\epsilon$, $y = y'\epsilon$, and $z = z'\epsilon$, where t' , x' , y' , and z' are dimensionless variables. We finally observe that the relevant parameters that describe the system dynamics are the dimensionless advection velocity, $v' = \frac{v}{M\kappa_B\epsilon}$, the dimensionless repulsion coefficient, $\gamma' = \frac{\gamma}{\kappa_B}$, and the dimensionless adhesion coefficient, $\eta' = \frac{\eta}{\kappa_B\epsilon^2}$. Nevertheless, in this work we will be using equation (2) in its dimensional form for a more direct substitution of the experimental parameters.”

7: The numerical calculation method seems to be almost appropriate. On the other hand, I didn't quite understand how to handle the advection term in the second term on the left side of equation (2). Advection terms can easily lead to numerical instabilities, so the author used some method (upwind differencing scheme??), didn't he? Please mention this.

Advection terms in phase-field often create numerical problems when the velocity depends on the free energy functional through a Navier-Stokes equation (in model H, according to the Hohenberg-Halperin classification) or through a force balance equation that includes a drag term. In our model, however, the velocity is set to a constant value (in space) for each cell (that produces the translation movement of the erythrocytes against each other), and we find that the simulation time step is limited by the fourth derivative of the order parameter in equation (2), and not by the advection term. Hence, we did not observe any instability resulting from the velocity term, and our semi-implicit method was able to handle the system of equations in 3D, as described in the numerical methods section of the manuscript.

REVIEWERS' COMMENTS:

Reviewer #1 (Remarks to the Author):

Nice paper, the authors have provided a detailed response to all the comments and have further improved the manuscript.

Reviewer #2 (Remarks to the Author):

The authors addressed my comments. I recommend publication.

Reviewer #3 (Remarks to the Author):

Thank you for your detailed comment.
I have no questions.
Thank you very much.